# Metal Organic Frameworks: Current State and Analysis of Their Use as Modifiers of the Vulcanization Process and Properties of Rubber

**DOI:** 10.3390/ma16247631

**Published:** 2023-12-13

**Authors:** Katarzyna Klajn, Tomasz Gozdek, Dariusz M. Bieliński

**Affiliations:** Institute of Polymer & Dye Technology, Lodz University of Technology, 90-537 Lodz, Poland; katarzyna.klajn@dokt.p.lodz.pl (K.K.); tomasz.gozdek@p.lodz.pl (T.G.)

**Keywords:** MOF, rubber, vulcanization, crosslink structure, thermal properties, mechanical properties

## Abstract

The interest in and application of metal organic frameworks (MOF) is increasing every year. These substances are widely used in many places, including the separation and storage of gases and energy, catalysis, electrochemistry, optoelectronics, and medicine. Their use in polymer technology is also increasing, focusing mainly on the synthesis of MOF-polymer hybrid compounds. Due to the presence of metal ions in their structure, they can also serve as a component of the crosslinking system used for curing elastomers. This article presents the possibility of using zeolitic imidazolate framework ZIF-8 or MOF-5 as activators for sulfur vulcanization of styrene-butadiene rubber (SBR), replacing zinc oxide in conventional (CV) or effective (EF) curing systems to different extents. Their participation in the curing process and influence on the crosslinking density and structure, as well as the mechanical and thermal properties of the rubber vulcanizates, were examined.

## 1. State-of-the Art in the Field of MOFs

Coordinate polymers, the term first used in 1916, are compounds with a structure formed by metallic ions and organic and/or inorganic ligands that are bound via a coordinate bond. This connection can lead to one- two- or three-dimensional networks [1,2]. One of the attempts to classify these compounds resulted in their division into three groups [3]:the first group—materials with a system of internal openings filled and supported by guest particles, which is destroyed when they leave,the second group—porous materials with a strong skeleton system that persists even after the removal of guest particles,the third group—materials with a flexible, pliable skeleton that can change reversibly under the influence of guest particles or in response to external stimuli such as light, temperature, or an electric field. Other names for this group are dynamic porous coordination polymers [4] or breathable materials [5].

### 1.1. Characteristics of MOFs

According to the above classification, MOF (metal organic framework) materials belong to the second or the third group of coordination polymers. The term MOF was introduced by Yaghi in 1995 [6]. The coordinating connection creates organized and repeating three-dimensional elements that form the crystal lattice, characterizing itself by an ordered system of pores resulting from the specific bonds between the metal and organic ligands. They are mostly microporous materials, but some of them have larger sorption holes (mesopores). MOFs are characterized by record-high specific areas, e.g., MOF-200–6200 m^2^/g [7]. Due to their huge specific surface areas, some researchers question the use of standard research methods and their correctness in studying MOFs [8]. MOF materials show large pore volumes, even above 2 cm^3^/g. The property distinguishing MOF materials from other porous solids is the flexibility of their structure, a result of the possibility of changing the shape of their skeleton [9]. An important property is also the shrinkage of the MOF material skeleton with increasing temperature. Some MOF materials (e.g., MIL-88 [10]) exhibit the ability to swell when exposed to the presence of a guest molecule (e.g., originating from a solvent), which causes significant changes in their structure. Many MOF materials have two or more permanent crystallographic structures (e.g., IRMOF-1 and MIL-53 [11]), which vary depending on conditions such as the temperature, pressure, and the presence of guest particles.

Metals used for the synthesis of MOFs are mainly cations of transition metals, but also alkaline earth metals, metals of the main groups of the periodic table, or rare earth metals. When designing the geometry of MOFs’ structures, the following parameters should be considered, namely: the type of element used, its valence, and also the typical coordination numbers assumed by a given element [4,8,12,13,14]. On the other hand, organic ligands should contain electron donors in their structure. In the case of nitrogen-containing compounds, substrates with amide, imidazole, pyridyl, cyano groups etc. can be used. When the electron donor is oxygen, aromatic oligocarboxylic molecules and also molecules containing phosphonium or sulfone groups can be applied. Cationic ligands are used less frequently due to their lower coordination affinity for metal cations. An important parameter of the organic ligands is their geometry and length, which generate the size of the pores [4,15,16].

### 1.2. Synthesis of MOFs

There are several known methods for synthesizing MOFs: solvo/hydrothermal, microwave, ultrasonic, electrochemical, or mechanochemical. These techniques allow the processes to be run under mild conditions and with the use of inexpensive precursors [17,18].

A typical environment for the synthesis of MOF materials is the liquid phase. These syntheses are based mainly on mixing two solutions containing a metal precursor and an organic ligand. Then, the solvothermal (hydrothermal) process is carried out at an elevated temperature. In addition to the basic substrates, agents that direct or aid crystallization, as well as agents that modify the reactivity of the mixture, can be added to the reaction mixture. The method of carrying out the synthesis (mixture composition, the order of adding components, solvent, pH, crystallization time, temperature, etc.) can have a large impact on its result. Seemingly trivial differences in the synthesis method may lead to the production of various compounds [12,16]. A slight change in the metal to ligand ratio may result in obtaining different structures. It is possible, inter alia, to obtain seven different zinc imidazole MOF structures from the same initial mixture using different solvents [19]. A similar effect is observed with the use of zinc and terephthalic acid to obtain various structures: MOF-2, MOF-3, or MOF-5 depending on the solvents used [20].

It is also possible to obtain MOF materials without any solvents by tribochemical methods. They involve the use of mechanical energy generated during the grinding of substrates to carry out the resulting chemical reactions to form MOF materials. The great advantage of these techniques is the small volume of the reaction mixture, the elimination of solvents, and the reduction of post-reaction waste. However, these methods are not universal, and their effectiveness has been described only for a few materials [21,22]. There is also an electrochemical method, patented by BASF, which enables the preparation of e.g., ZIF-8 material [23,24].

Additionally, microwave radiation and ultrasound can have a positive effect on the course of the synthesis of MOF materials. The advantage of these methods is their short crystallization time and the ability to control the size and shape of the pores formed in the material by the appropriate selection of the synthesis conditions [25,26].

MOF materials are inherently insoluble, which makes it impossible to use traditional methods of purifying organic compounds, such as distillation, recrystallization, chromatography, or sublimation. Obtaining pure materials can be achieved by optimizing the synthesis conditions (such as the composition of the solvents, concentration of reagents, reaction time, and even the size of the reactor), which entails the modification of many variables. It is often possible to manually select the desired crystals (if their color or shape is clearly different from the impurities). A faster gravimetric purification method applies solvents of different densities. This method allows for the separation of a MOF mixture that differs only in the arrangement of metallic nodes and organic bridges in space, and it also allows the separation of MOFs with one type of organic ligand from MOFs made up of a mixture of ligands. It is also possible to separate MOF structures, the skeletons of which are intertwined with each other, from those that do not exhibit such an interleaving [27].

### 1.3. Modification of MOFs: Possibilities and Limitations

For most applications in which MOFs can be used, it is necessary to remove the solvent particles from the pores of the MOF materials. Traditional activation requires heating MOF materials in a vacuum, leading to partial or complete loss of their porosity. To avoid this problem, it is possible to replace the solvent remaining in the pores after synthesis with a solvent with a lower boiling point, thanks to which its activation can take place under milder conditions [28]. An alternative way to perform the activation is to use supercritical CO_2_ liquid [29].

The first option to modify MOF particles is doing it during their synthesis. It is possible to use an organic ligand that contains additional functional groups that are not involved in the formation of a MOF structure [30]. In this strategy, it is possible to use all ligands containing functional groups or functional mixtures with non-functional ones, thanks to which various structures can be formed. Another strategy uses metallic ligands that contain a metal that does not participate in the formation of the knot in the skeleton of the resulting material, but it may affect its catalytic properties. It is also possible to use mixed metal cations. For a typical synthesis, the second type of metal cation is added, which is to be included in the metal cluster [31,32].

On the other hand, modification can take place when the synthesis is completed. It most often includes the: exchange of guest particles, removal of guest particles, ion exchange, or encapsulation of nanoparticles. Removal of the guest particles from the interior spaces of MOFs can result in slight structural changes, which are usually reversible. Modifications of their properties can occur because of these changes. The process may be reversible, and the material may absorb compounds such as: pyridine, benzene, or nitrobenzene [7,33,34]. Due to these properties, MOFs show similarity to zeolites. An additional similarity is the possibility of exchanging ions inside these materials, with the difference that MOFs can undergo both anionic and cationic exchange, depending on the charge of the backbone [35]. It is also possible to introduce into their interior highly dispersed metal particles, such as Pt, Au, Pd, or Ru, using the CVD (chemical vapor deposition) technique or wetting impregnation [36,37,38].

Modification of MOF materials can be distinguished using coordination interactions with metals forming the structure. The first is the introduction of organic molecules into the exposed coordination sites of metals that make up the skeleton. It has been proven that water molecules located near metal centers can be replaced with other molecules, and additionally, treating such material with pyridine allows for the creation of a new MOF structure. It is also possible to treat the dehydrated material with multifunctional organic amines, making the obtained materials even more reactive [39,40,41,42,43]. Another possibility for modification is to use unbound functional groups of the incoming organic ligand as part of the MOF backbone for the formation of a coordinate bond with the introduced modifying molecule [44,45].

Covalent bonds are also used for modification, but due to their strength, this is possible only in some cases. For MOFs containing amine or formyl groups in the wall structure, they can be used for: amide coupling [46,47], imine condensation [48,49], urea formation [50], bromination [47], reduction [48], or modification with the use of acid anhydride, and bridging modifications of the hydroxyl group located on metal cations have also been described [51].

It is important to keep in mind the limitations resulting from the porous structure of the material. The pore size determines the maximum size of the guest molecule, making it a perfect match for the material being modified. Another important condition is the stability of the modified structure, its sensitivity to temperature, and its relatively low chemical stability, which depends on the reaction conditions [52].

It is also possible to create hybrid structures, one part of which is a carbon carrier, including e.g., carbon nanotubes. The structure of CNT-MOF is of great interest, as it allows for an increase in thermal and chemical stability as well as improved adsorption and mechanical properties. For the synthesis, the CNTs are dispersed in a solution of MOF precursors [53]. It is also possible to use the MOF material—graphene hybrid, which significantly increases the possibility of gas absorption, while eliminating the disadvantages of MOFs such as: slow decomposition at room temperature and limited resistance to water and temperature [54].

### 1.4. Application of MOFs

Along with the ability to control and modify the resulting structures, which leads to obtaining unique, multifunctional, and unconventional physicochemical properties, the characterization of MOFs has opened new possibilities for their use not only in many branches of chemistry, but also in related disciplines, such as materials engineering, nanotechnology, physics, energy, biology, medicine, and environmental engineering [18,55]. They can be used in many fields, including [18,56]:gas storage, separation, and purification [57,58,59,60],energy storage [61],heterogeneous catalysis [62,63,64,65],electrochemistry (from batteries, through supercapacitors, to fuel cells) [66,67,68],optoelectronics [69],luminescent materials [70,71,72,73],medical applications (drug delivery, diagnostic tests, and imaging) [74,75,76,77], andchemical sensors [10,73].

MOF materials were tested on a larger scale, e.g., during a car trip through Asia—from Berlin to Bangkok (2007). The car (Volkswagen Caddy—Volkswagen, Germany, Eco Fuel—BASF, Germany) was fueled with natural gas stored in a MOF material (Basolite C300). The tested material was able to hold 30% more fuel, enabling a 20% increase in range between refuelings [78]. Another example of the practical use of MOF materials is the absorption of ethylene (in closed packages) released by ripening fruit during transport [11].

#### Polymers Modified with MOFs

MOF—polymer systems can often be used in similar applications as the MOF materials themselves, but with additional advantages over polymers alone concerning adsorption, capture, or degradation of compounds. The combination of the biodegradable polymer Ecoflex and Basolite M050—a MOF material based on magnesium and formic acid—was intended to produce packaging that would effectively absorb ethylene and remaining biodegradable [79].

One of the possibilities of creating a polymer/MOF hybrid is polymerization in the holes. The process can be carried out by simultaneous growth of the polymer and the MOF material, or an appropriate polymer/MOF can be used as a matrix, with which the remainder is co-synthesized. For example:PS polymerization leading to the creation of homogeneous chains without interactions between them, with a low activation energy and elimination of the glass transition temperature [80];PEG polymerization allowing for a reduction in the temperature of the thermal changes, which is important due to the specific structure of MOFs. It is possible to separate chains according to the end groups, which differ not only in their elemental composition, but also in polarity [81,82].

In addition, the polymers present in the MOF pores affect the mechanical properties of the material, increasing its compression strength, as well as affecting the structure and deformation of the pores, especially for elastomers [83,84].

Organometallic compounds can also be used to synthesize complex architectural forms or molecular wires to cope with the strong interactions between individual strands [85]. The materials synthesized by this method, apart from having a specific shape, can also have designed properties. It is possible to create composites used as solvent-applied films, e.g., a mixture of HKUST-1 MOF and poly(3-hexylthiophene) (P3HT) used in thin film transistors [86] or sulphonated polythiophene with a zinc-based MOF and meso-tetra-(4-carboxyphenyl) porphyrin for use in a solar cell, where the MOF material is responsible for the initiation of the reduction process [87]. Likewise, it is possible to use the flexibility of silicone rubber or styrene-butadiene copolymers for the formation of composites containing MOFs to be used as membranes. What is also important is that MOFs can improve the thermal stability and thermal aging resistance of silicone rubber [88]. The formation of the composite may take place by applying the polymer to the finished membrane or by creating a membrane of a dissolved polymer and a MOF material dispersed therein [89,90]. Such composites can also be used in 3D printing, functionalizing products due to the large specific surface of MOFs. Printed ABS-MOF-5 composites could be used for capturing hydrogen. Thanks to the use of the additive, the gas storage capacity of ABS was significantly increased while maintaining the mechanical properties of the polymer [91].

Another approach is to apply a layer containing MOF to the finished, printed element, e.g., to create a dye adsorbing layer [92]. There are many similar approaches to the use of this type of system, mainly for gas adsorption, liquid purification, or as catalysts, e.g., MOF-PVA [93] or HKUST-1-methacrylates [94]. By using the properties of MOF applied to the surface of the polymer, it increases its ability to absorb moisture or react with light. The materials produced in this way can be used, for example, as sensors that respond to the environment [95,96]. Due to the reversibility of changes in the structure of organometallic compounds, it is possible to produce self-healing gels containing, e.g., ZIF-8, ZIF-67, or UiO-66 MOFs [97]. MOF structures can be synthesized on various surfaces, including, for example, cellulose [98], but also nylon fibers using the amino group (UiO-66) [99]. The materials made each time show a significant change in their properties, significantly widening their possible applications in composites [86,100].

Organometallic compounds have also been used as components of rubber mixtures. Their application has been patented in pneumatic tires containing at least one diene rubber and one MOF containing Zn^2+^ ions and a multifunctional ligand in the amount of 0.1–50 phr, as well as silica or carbon black. This additive may influence the crosslinking of the material and its ultimate properties [101].

As importantly, it is speculated that MOF can also improve the thermal stability and thermal aging resistance of silicone rubber due to reduced heat transport and blocking of polymer chain degradation [88]. Attempts have also been made to use organometallic compounds containing aluminum (as a metallic cluster) and 2-aminoterephthalic acid (as an organic ligand) as a filler for SBR rubber. Studies have shown that, relative to unfilled vulcanizate, samples made of MOF have significantly better mechanical properties, which are comparable to composites containing nanofillers, such as nanorosilica or nanotita-nium [102]. Other studies using styrene—butadiene rubber have shown that it is possible to use MOF systems containing zinc clusters as activators of vulcanization. In their work, these researchers used an organometallic compound deposited on the surface of microme-ter-sized zinc oxide particles, which enabled better dispersion of the MOFs, improving the crosslinking density and mechanical properties of the vulcanizates. It should be noted, however, that the amount of zinc present in the entire mixture was summed to be higher than in the reference sample, which contained only zinc oxide [103]. The addition of three different zeolitic imidazolate frameworks to natural rubber-based blends as a functional additive allowed, with a suitable accelerator system, a reduction in the vulcanization temperature while maintaining or improving its mechanical properties and increasing its crosslinking density [104]. These relationships resulted from the higher Zn content in the blends. The addition of ZIF also resulted in increased aging resistance of the compositions.

## 2. Introduction to Original Own Research

Inspired by the obtained effects [88,101], we decided to investigate the effectiveness of MOFs containing zinc ions, introducing them to a sulfur curing system, from the point of view of their influence on the kinetics of crosslinking, crosslink density, and structure, as well as the related thermal and mechanical properties of styrene-butadiene rubber vulcanizates (SBR). Vulcanization involving MOFs, the crosslink density and structure, and the selected properties of SBR were compared to rubber cured with conventional and effective sulfur curing systems.

## 3. Materials

### 3.1. MOF Structure and Zinc Amount

Commercial compounds from the MOF group were used for this research: ZIF-8 (MOF Technologies Ltd., Belfast, UK) and MOF-5 (novoMOF AG, Zofingen, Switzerland). According to their producers, the first one is uncontaminated, while the second one is 99% pure. Rubber mixtures were prepared keeping the same amount of zinc in the composition as in the reference samples containing ZnO as an activator of sulfur vulcanization (ZnO has 80% of the zinc in its structure, so its 3 phr addition is an equivalent of 2.4 phr of Zn^2+^ from MOFs). Knowing that the amount of Zn in the ZIF-8 structure (Figure 1) is 28.7%, the equivalent replacement of 100% ZnO in the reference rubber mixture requires the addition of 8.4 phr of ZIF-8. MOF-5 contains 34.0% Zn in its structure, so its full equivalent of ZnO is 7.0 phr.

### 3.2. Preparation of Rubber Samples

The compositions of the rubber mixtures studied are presented in Table 1. Codes of the samples include: the abbreviation of the crosslinking system used (CV—conventional, EF—effective), and the zinc content, taking into consideration the amount of zinc oxide and the metal organic frameworks in the percent share (e.g., CV_25_ZnO_75_MOF-5).

All of the ingredients were mixed using a David Bridge laboratory scale two rolls mill (David Bridge & Co., Rochdale, UK). The rubber vulcanizates were prepared by steel molding using a laboratory hydraulic press operating under a pressure of 200 bar and at a temperature of 160 °C, during an optimum vulcanization time (t_90_) determined rheometrically with a MDR 2000 vulcameter (Alpha Technology, Hudson, OH, USA), according to PN-ISO 3417:2015-12.

## 4. Experimental Techniques

### 4.1. Kinetics of Vulcanization

Vulcanization kinetics of the rubber mixtures were analyzed using an Alpha Technologies MDR 2000 oscillating disk rheometer (Alpha Technologies, Hudson, OH, USA) at 150, 160, and 170 °C, according to PN-ISO 3417:2015-12. The following curing parameters were determined based on the experimental data: optimum vulcanization time (t_90_), vulcanization scorch time (t_s2_), max. torque (M_H_), and min. torque (M_L_). The increase of torque was calculated as ΔM = M_H_ − M_L_. The conventional cure rate index (CRI) of the rubber mixtures was calculated according to Equation (1) [105,106], as follows:(1)$$CRI=\frac{100}{t_{90}-t_{s2}}$$

Based on the vulcametric data, the vulcanization kinetics of the rubber mixtures were characterized in terms of the activation energy (E_a_), calculated according to the Arrhenius formula—Equation (2):(2)$$\ln k(T)=\ln A-E_{a}/(RT)$$

The rate constant (k) of the reaction was calculated using non—linear regression, according to the Kamal–Sourour [107] model—Equation (3):(3)$$d\alpha /\mathrm{dt}=1/(M_{H}-M_{L})\cdot dM/dt$$
where:

α(t) is the degree of vulcanization at a given time and M(t) is the torque at a given time.

This enabled the course of vulcanization rate (dα/dt) to be determined as a function of the degree of vulcanization (α).

### 4.2. XRD of MOF Containing Samples

Room temperature powder X-ray diffraction patterns were collected using a X’Pert Pro MPD diffractometer (Malvern Panalytical Ltd., Royston, UK) in the Bragg–Brentano reflection geometry. Copper CuKα radiation was used from a sealed tube. Data were collected in the 2θ range 3–70° with a step of 0.0167° and an exposure per step of 20 s. The samples were spun during data collection to minimize preferred orientation effects. A PANalytical X’Celerator detector based on Real Time Multiple Strip technology and capable of simultaneously measuring intensities in the 2θ range of 2.122° was used. The XRD spectra of the virgin ZIF-8, rubber mix, and vulcanizate containing the MOF were compared in order to check their structural stability during processing.

### 4.3. ToF-SIMS of MOFs

This research used a TOF-SIMS IV secondary ion mass spectrometer from ION-TOF GmbH (Münster, Germany), equipped with a bismuth ion gun (Bi3^+^) and an ion time-of-flight analyzer with high mass resolution m/Δm = 7500 for an ion with *m*/*z* equal to 29. Samples were made by pressing MOF and MOF/stearic acid powders. During the measurement, the emission of secondary ions was obtained as a result of irradiating the surface of the tested sample with a pulsed beam of primary ions. The frequency of the primary ion beam was 10 kHz, and the duration of a single pulse was approximately 1 ns. The average primary beam current was 0.2 pA at an ion energy of 25 keV. Secondary ion mass spectra were recorded from a surface region of dimensions 100 × 100 μm. The number of characteristic ions containing zinc was counted, comparing their number between virgin MOFs and MOF samples annealed with an appropriate amount of stearic acid (according to the recipe of a rubber mixture) under vulcanization conditions (160 °C, 40 min).

### 4.4. Crosslink Density and Structure of the Rubber Vulcanizates

The equilibrium swelling of the rubber vulcanizates in toluene (Chemia Lodz, Poland) was determined according to the standard procedure, available in the literature [108]. The crosslink density of the vulcanizates was calculated from their volumetric equilibrium swelling values, applying the Flory–Rehner equation [109] with an SBR–toluene Flory-Huggins’ parameter of 0.378 [110]. The structural composition of the sulfide crosslinks was evaluated according to the procedure described by Saville and Watson [111], based on selective dissolving of rubber vulcanizates in thiol-amine solvents (OTAM/OTAT), depending on the length of the crosslinks.

### 4.5. Thermal Properties of the Rubber Vulcanizates

The thermal stability of the rubber vulcanizates was studied using a Mettler Toledo TGA instrument (Mettler Toledo, Columbus, OH, USA), operated in a temperature range from room temperature to 800 °C with a heating rate of 10 deg/min. The sample mass was about 10 mg. The temperature of 50% mass loss of the samples (T_50_) was determined and proposed as an indicator of their thermal stability.

The glass transition temperature of the rubber vulcanizates (T_g_) was determined using a Mettler Toledo DSC 1 instrument (Mettler Toledo, Columbus, OH, USA) operated in a temperature range from −100 to 30 °C with a heating rate of 10 deg/min. The sample mass was also about 10 mg.

### 4.6. Mechanical Properties of the Rubber Vulcanizates

The mechanical properties of the rubber vulcanizates were determined using a Zwick 1435 universal mechanical testing machine (Zwick Roell GmbH & Co. KG, Ulm, Germany), according to ISO 37. Dumbbell-shaped specimens of ca. 2 mm thickness were used. Five samples of each material were analyzed and the experimental data averaged.

## 5. Results and Discussion

The addition of MOFs to the rubber mixtures changed the nature of the vulcametric curve to a stepping one, together with lowering of the maximum vulcametric torque. In addition, in the case of a small addition (max. of 50% zinc originated from MOF), it decreased the optimum crosslinking time (Table 2). The rate of vulcanization for samples containing ZIF-8 decreased with its increasing content in the mixtures, regardless of the crosslinking system used. Despite the effect being analogous to MOF-5 and an effective crosslinking system, for the conventional one, the highest rate of vulcanization was observed for the samples containing 25% and 100% Zn originating from MOF.

This study showed that activation of the vulcanization process in a conventional curing system with zinc oxide as an activator requires energy of less than 40 kJ/mol. Using only a ZIF-8 equivalent as an activator, instead of ZnO, resulted in a slight decrease in the activation energy (E_a_) of sulfur vulcanization. However, replacing only 25% of the zinc source with ZIF-8 resulted in a significant increase of E_a_, and the start of the process required as much as 1.5 times more energy. It is noteworthy that despite this, the entire vulcanization process was quite fast, being probably sufficient in terms of energy after crossing a rather large initial barrier. As the amount of the ZIF-8 additive increased, the activation energy of vulcanization decreased. The complete replacement of zinc oxide with MOF-5 in a conventional curing system resulted in significant increase in the activation energy of vulcanization. The increase in energy with the addition of 25% Zn from ZIF-8 may be a result of the introduction of larger particles into the system, disturbing the action of ZnO, and due to their relatively small share, their action as an activator also requires quite large energy inputs. As the amount of ZIF-8 added increased, the vulcanization activation energy decreased. Thus, for the CV_50_ZnO_50_ZIF-8 sample, it was lower than for the reference sample, and the effect increased even more if we consider the sample containing 75% zinc from ZIF—8. For this best sample, the activation energy dropped below 20 kJ/mol. Probably, in these systems we are dealing with synergy of action of both activators, which start working relatively quickly, and starting the reaction does not require large energy expenditure. A sample containing 100% of zinc originating from MOF—5 in a conventional curing system required the highest energy expenditure among all of the rubber mixtures tested. Using MOF-5, however, the synergistic effect of the two activators (ZnO and MOF-5) can be observed. Samples containing 25% or 50% of zinc originating from MOF-5 require equal energy inputs, and in addition, the value of the activation energy of the curing process is perfectly in the middle between the reference sample (100% ZnO) and the sample containing only MOF-5 as an activator. The role of ZIF-8 and MOF-5 is to activate the vulcanization process by donating zinc.

XRD studies confirmed the stability of ZIF-8 after 24 h at 200 °C in the presence of air. Its thermal decomposition, determined using TGA, also began far above the tempera-ture of rubber vulcanization [112]. Similarly, for MOF-5, the compound decomposed only above 400 °C [113]. We do not have confirmation that the zinc ions remained 100% in place after the vulcanization process, but the XRD spectrum still showed peaks characteristic of ZIF-8—Figure 2. Their lower intensity results from the fact that we are examining ZIF-8 crystals in rubber. The authors of the article [112] indicate the presence of bands characteristic of ZnO are formed after the decay of ZIF-8. This effect was not observed in our vulcanizates.

Comparison of the ToF-SIMS spectra of both MOFs before and after heating at the vulcanization temperature indicates the possible release of part of the zinc from their structure. The MOF-5 sample is definitely less homogeneous than the ZIF-8 sample in terms of zinc content. For the former, ions with organic ligands, like 
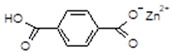
, were expected to be emitted, if they are formed at all. Unfortunately, neither such ions nor ions containing zinc and other organic ligands with larger masses were observed in the spectra.

However, there is a peak in the spectrum of the ZIF-8 sample that can be assigned to an ion containing zinc and the methylimidazole ligand (C_4_H_5_N_2_Zn^+^).

Much more “interesting” are the spectra of MOFs samples with the addition of stearic acid as a sulfur vulcanization activator, subjected to heating under vulcanization conditions (temperature and time)—Table 3.

It can be noticed that as a result of subjecting MOFs samples with stearic acid to heating under vulcanization conditions (160 °C, 40 min), ions containing zinc and stearic acid (C_18_H_36_O_2_) appear in the ToF-SIMS spectra. This confirms the participation of MOFs in the rubber vulcanization process, indicating the participation of MOFs in the sulfur curing of rubber. The observed effect is greater in relation to ZIF-8.

In the effective crosslinking system, the initiation of the vulcanization reaction requires slightly higher energy inputs (Table 2). As in the conventional system, also in this case, replacing all of the zinc oxide with ZIF-8 resulted in a decrease in the activation energy, but in the case of an effective crosslinking system, the effect obtained was higher. Therefore, it can be concluded that the introduction of ZIF-8 has a positive effect on this parameter. In addition, a synergistic effect of both activators added in a ratio of 1:1 was observed, for which the activation energy of the vulcanization process was the lowest. The rubber mixture with the highest activation energy was EF_0_ZnO_100_MOF-5. Probably, this particular MOF requires much more energy supplied to the system to be able to adequately bring about the crosslinking of rubber. Despite the activation energy of vulcanization for the other rubber samples containing a mixture of the activators, they are similar to each other, but are slightly higher for the sample in which the ratio of ZnO to MOF is 1:1, where the synergism of the additives’ effect is apparent.

The addition of MOFs in each case increased the percentage of polysulfide crosslinks in the vulcanizates, but the effect was different (Figure 3). As expected, the highest contents of mono— and disulfide crosslinks were obtained for the rubber samples cured with an effective sulfur system (EF), no matter the ZIF-8 or MOF-5 application. However, the zinc source in the form of MOF-5 resulted in a higher crosslink density of rubber in comparison to ZIF-8. This can be explained by the presence of carboxylic groups in the latter, facilitating the formation of a zinc/carboxylate complex during the vulcanization reaction [114].

The highest crosslink density for the rubber mixtures cured with the addition of ZIF-8 can be observed for 75_ZnO_25_ZIF-8 vulcanizates, no matter in the conventional or effective crosslinking system. Contrary to this, for MOF-5 containing mixtures, the crosslink density increased with increasing content of the MOF only in the case of the conventional crosslinking system, decreasing for the effective one (Figure 3). However, even for the EF_0_ZnO_100_MOF-5 sample, its density is significantly higher in comparison to the reference sample.

Whenever the samples are stretched, the mechanical properties of the vulcanizates are affected not only by the density of the crosslinking, but also by the structure of the crosslinks (Figure 4). Often, it is only one of these parameters that determine the properties of the resulting vulcanizates and on it, in the main, depends the mechanical characteristics of the materials. The reference sample, crosslinked with a conventional crosslinking system, is characterized by a rather low elasticity—the value of its relative elongation at break does not exceed 300%. Similar to it, the sample containing 100% zinc derived from ZIF-8, is also stiff. Despite having the lowest crosslink density, it is its content of short bonds that in this case influence the stresses at given elongations higher than the reference. Vulcanizates containing both crosslinking activators (ZnO and ZIF-8), allow for higher relative elongations in tension, each time above 300%. In their case, the higher elasticity of the vulcanizates is most likely due to the high content of polysulfide crosslinks in the structure of their spatial network. Vulcanizates crosslinked with a conventional curing system containing MOF-5 are more similar, according to their mechanical properties, to the reference sample, but they are more elastic, which can also be related to the higher presence of polysulfide crosslinks in the spatial network. Using an effective crosslinking system with ZnO as an only activator, the reference sample elongates during stretching by more than 300%. It is important to note that the ZnO-free sample, containing only ZIF-8, has slightly higher mechanical modulus values than the reference sample, despite the lower crosslink density. Similar to the conventional system, the values of stresses at a given strain for the samples containing both activators (ZnO and ZIF-8) arrange parabolically, with the maximum falling on EF_50_ZnO_50_ZIF-8.

Stresses at increasing elongations for the samples containing MOF-5 cured with an effective crosslinking system differ slightly from the reference sample. However, each time, a slight increase in stress can be observed, which is most pronounced for the sample containing ZnO and MOF-5 in a ratio of 1:1. For this proportion, the most favorable ratio of crosslinking density (the second highest) to the content of polysulfide crosslinks is generated, explaining the high strength and elasticity of the vulcanizates.

The modification of the conventional crosslinking system by the addition of ZIF-8 had no significant effect on the thermal decomposition of the rubber vulcanizates studied, represented by T_50_ (Table 4). It caused only a slight decrease in their glass transition temperature, which, however, does not depend directly on the amount of the MOFs used. Additionally, the differences between samples are minor, so it can be concluded that the addition of the MOF does not adversely affect the thermal stability of the vulcanizates. Contrary to the above, in the case of crosslinking systems containing MOF-5, regardless of the type, a reduction in the temperature of 50% material decomposition was observed.

## 6. Conclusions

Metal organic frameworks (MOFs) of cage structures are an interesting and versatile group of chemicals, which are used in many applications, both in the laboratory and recently more and more often on the industrial scale [18]. The presented review discusses the use of MOFs for the separation and storage of gases and energy, catalysis, electrochemistry, optoelectronics, medicine, and most recently, in polymer technology. Against this background, the application of this type of material for the modification of sulfur curing systems in rubber vulcanization was studied. The partial replacement of ZnO by the selected MOFs with zinc ions in their structure makes some changes to the vulcanization rate (CRI), its activation energy, and kinetics. The most optimal arrangement seems to be a 1:1 ratio of the activators. This results in changes to the crosslink density and structure of rubber (SBR) vulcanizates, especially when crosslinked with a conventional sulfur curing system, which affects their mechanical properties. No significant changes to the thermal stability or the glass transition temperature of the vulcanizates studied was recorded.

The results of the preliminary studies encouraged us to undertake further research on the use of MOFs in other aspects of rubber technology and exploitation, trying to exploit their large specific surface area and/or structural porosity.

## Figures and Tables

**Figure 1 materials-16-07631-f001:**
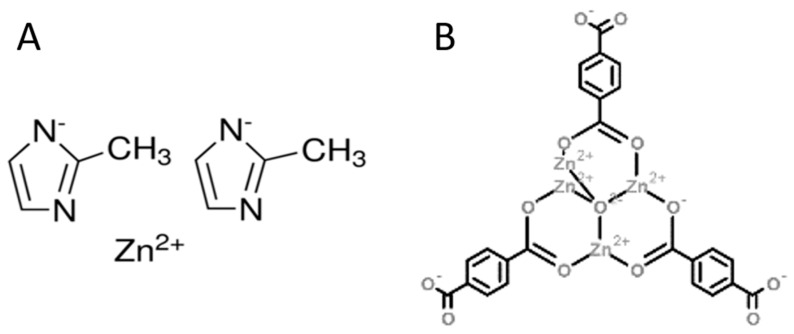
Structure of (**A**): ZIF-8 and (**B**): MOF-5.

**Figure 2 materials-16-07631-f002:**
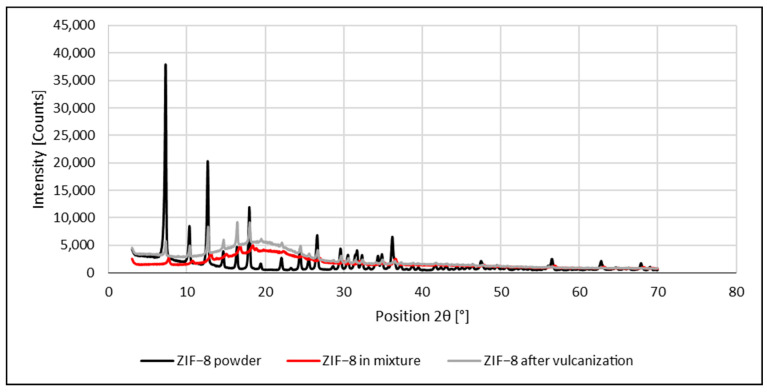
Comparison of XRD spectra of ZIF-8, ZIF-8 incorporated in the rubber mixture 0_ZnO_100_ZIF-8 (see Table 1), and ZIF-8 present in the rubber after vulcanization.

**Figure 3 materials-16-07631-f003:**
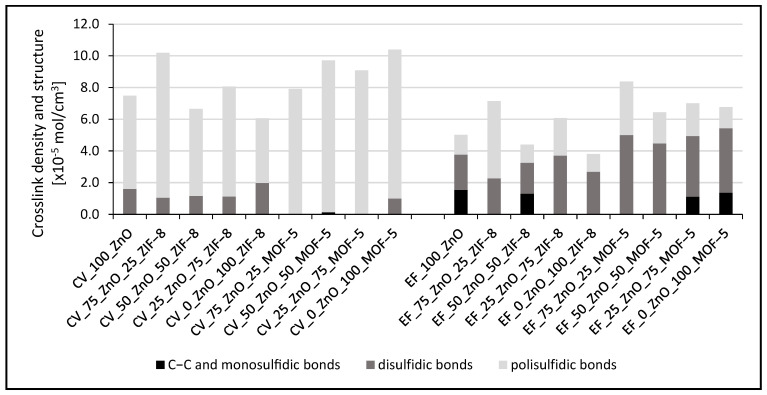
Crosslink density and structure of the rubber vulcanizates studied.

**Figure 4 materials-16-07631-f004:**
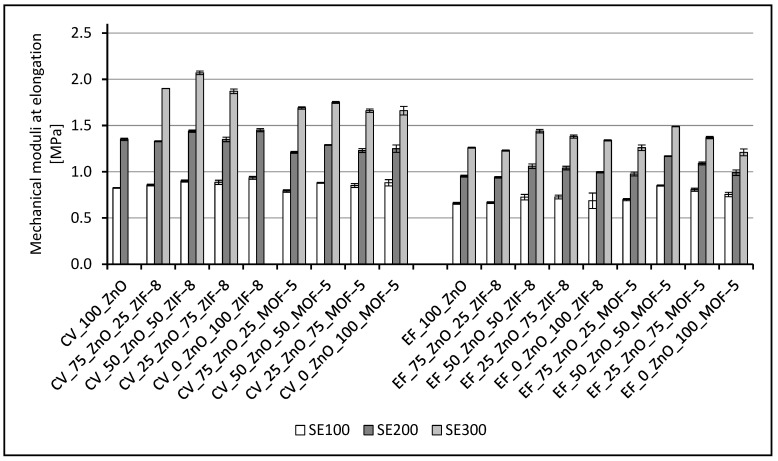
Mechanical properties of the rubber vulcanizates studied.

**Table 1 materials-16-07631-t001:** Composition (phr) of the rubber mixtures studied.

Components	Conventional Curing System (CV)								
	ZnO	75_ZnO_25_ZIF-8	50_ZnO_50_ZIF-8	25_ZnO_75_ZIF-8	0_ZnO_100_ZIF-8	75_ZnO_25_MOF-5	50_ZnO_50_MOF-5	25_ZnO_75_MOF-5	0_ZnO_100_MOF-5
**SPRINTAN^TM^ SLR-4602 (s-SBR)**	100.00								
**Sulfur**	3.00								
**N-cyclohexylbenzothiazol-2-sulphenamide (CBS)**	1.50								
**Stearic acid**	3.00								
**Zinc oxide (ZnO)**	3.00	2.25	1.50	0.75	-	2.25	1.50	0.75	-
**ZIF-8**	-	2.10	4.20	6.30	8.40	-	-	-	-
**MOF-5**	-	-	-	-	-	1.75	3.5	5.25	7.00
**Components**	**Effective curing system (EF)**								
	**ZnO**	**75_ZnO_25_ZIF-8**	**50_ZnO_50_ZIF-8**	**25_ZnO_75_ZIF-8**	**0_ZnO_100_ZIF-8**	**75_ZnO_25_MOF-5**	**50_ZnO_50_MOF-5**	**25_ZnO_75_MOF-5**	**0_ZnO_100_MOF-5**
**SPRINTAN^TM^ SLR-4602 (s-SBR)**	100.00								
**Sulfur**	0.60								
**N-cyclohexylbenzothiazol-2-sulphenamide (CBS)**	4.00								
**Stearic acid**	3.00								
**Zinc oxide (ZnO)**	3.00	2.25	1.50	0.75	-	2.25	1.50	0.75	-
**ZIF-8**	-	2.10	4.20	6.30	8.40	-	-	-	-
**MOF-5**	-	-	-	-	-	1.75	3.5	5.25	7.00

s-SBR—solution styrene-butadiene rubber.

**Table 2 materials-16-07631-t002:** Vulcanization parameters and activation energy of the rubber mixtures studied.

Sample	M_L_ [dNm]	M_H_ [dNm]	t_s2_ [min]	t_05_ [min]	t_90_ [min]	CRI [%/min]	E_a_ [kJ/mol]
**CV_100_ZnO**	0.18	9.50	9.10	8.16	20.02	9.,16	37.3 ± 1.1
**CV_75_ZnO_25_ZIF-8**	0.19	6.84	5.14	4.13	19.16	7.13	60.0 ± 2.2
**CV_50_ZnO_50_ZIF-8**	0.20	9.54	7.40	6.55	19.39	8.34	29.0 ± 0.9
**CV_25_ZnO_75_ZIF-8**	0.27	7.45	8.02	6.44	24.42	6.10	18.9 ± 0.6
**CV_0_ZnO_100_ZIF-8**	0.30	9.78	9.70	8.44	26.29	6.03	33.9 ± 1.2
**CV_75_ZnO_25_MOF-5**	0.25	7.06	10.31	8.67	18.78	11.81	62.8 ± 1.9
**CV_50_ZnO_50_MOF-5**	0.25	7.71	10.72	8.91	30.46	5.07	62.3 ± 2.2
**CV_25_ZnO_75_MOF-5**	0.27	7.71	14.33	10.23	34.10	5.06	30.0 ± 0.9
**CV_0_ZnO_100_MOF-5**	0.30	7.84	15.16	11.00	35.18	5.00	85.5 ± 2.6
**EF_100_ZnO**	0.19	6.67	16.09	13.59	24.54	11.83	46.2 ± 1.6
**EF_75_ZnO_25_ZIF-8**	0.20	4.63	13.87	10.03	20.10	16.05	46.6 ± 1.5
**EF_50_ZnO_50_ZIF-8**	0.21	6.10	12.08	8.37	22.12	9.96	28.3 ± 0.9
**EF_25_ZnO_75_ZIF-8**	0.25	5.27	20.03	13.62	30.92	9.18	83.3 ± 2.5
**EF_0_ZnO_100_ZIF-8**	0.22	7.01	19.71	15.50	35.09	6.50	33.5 ± 1.2
**EF_75_ZnO_25_MOF-5**	0.24	4.98	18.99	14.21	26.88	12.67	65.9 ± 2.0
**EF_50_ZnO_50_MOF-5**	0.22	5.43	29.49	12.13	42.07	7.95	71.1 ± 2.1
**EF_25_ZnO_75_MOF-5**	0.19	6.29	31.87	13.16	49.54	5.66	66.0 ± 1.9
**EF_0_ZnO_100_MOF-5**	0.21	6.52	34.80	13.99	69.48	2.88	103.7 ± 3.1

CV, EF—conventional and effective curing system, respectively; M_L_, M_H_—min. and max. vulcanization torque, respectively; t_s2_—the time from the start of the test to the moment when the torque value increases for 2 dNm above the M_L_ value; t_05—_scorch time (the time from the start of the test to the moment when the torque value reaches 5% of the M_H_ value); t_90_—optimum time of vulcanization (the time from the start of the test to the moment when the torque value reaches 90% of the MH value); CRI—Curing Rate Index; E_a_—activation energy of vulcanization.

**Table 3 materials-16-07631-t003:** Results of the FoF-SIMS tests of MOFs samples annealed with stearic acid.

**Positive Ions**				
	Zn^+^	C_4_H_5_N_2_Zn^+^	C_4_H_6_N_2_Zn^+^	C_18_H_36_O_2_Zn^+^
**ZiF-8**	+	+	+	-
**negative ions**				
	C_4_H_5_N_2_^−^	C_18_H_36_O_2_^−^		
**ZiF-8**	+	+		
**positive ions**				
	Zn^+^	C_8_H_5_O_4_Zn^+^	C_18_H_36_O_2_Zn^+^	
**MOF-5**	+	-	-	
**negative ions**				
	C_8_H_5_O_4_^−^	C_18_H_36_O_2_^−^		
**MOF-5**	+	+		

+/-: presence/absence of an ion peak in the sample spectrum.

**Table 4 materials-16-07631-t004:** Thermal properties of the rubber vulcanizates studied.

Sample	T_50_ [°C]	T_g_ [°C]	Sample	T_50_ [°C]	T_g_ [°C]
**CV_100_ZnO**	477	−15.5	**EF_100_ZnO**	485	−19.2
**CV_75_ZnO_25_ZIF-8**	474	−16.4	**EF_75_ZnO_25_ZIF-8**	478	−20.6
**CV_50_ZnO_50_ZIF-8**	476	−16.3	**EF_50_ZnO_50_ZIF-8**	483	−18.5
**CV_25_ZnO_75_ZIF-8**	474	−16.7	**EF_25_ZnO_75_ZIF-8**	479	−20.0
**CV_0_ZnO_100_ZIF-8**	477	−16.6	**EF_0_ZnO_100_ZIF-8**	485	−19.3
**CV_75_ZnO_25_MOF-5**	473	−16.8	**EF_75_ZnO_25_MOF-5**	480	−20.4
**CV_50_ZnO_50_MOF-5**	472	−16.7	**EF_50_ZnO_50_MOF-5**	479	−19.7
**CV_25_ZnO_75_MOF-5**	472	−16.9	**EF_25_ZnO_75_MOF-5**	478	−19.7
**CV_0_ZnO_100_MOF-5**	471	−16.7	**EF_0_ZnO_100_MOF-5**	478	−20.0

T_50_—temperature corresponding to 50% weight loss of the sample; Tg—glass transition temperature.

## Data Availability

Data available on request from the corresponding author.

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
