# Peer review of "Metal Organic Frameworks: Current State and Analysis of Their Use as Modifiers of the Vulcanization Process and Properties of Rubber"

_materials, 2023, doi:10.3390/ma16247631_

Round 1

Reviewer 1 Report

Comments and Suggestions for Authors

This paper discusses the potential use of metal-organic frameworks (MOFs) as modifiers of the vulcanisation process and rubber properties. It provides an overview of the growing interest and application of MOFs in various fields, including polymer technology, and explores the use of the zeolitic imidazolate framework ZIF-8 as an activator for the sulphur vulcanisation of styrene-butadiene rubber (SBR) and its impact on the cross-linking process, cross-link density and structure, as well as the mechanical and thermal properties of the vulcanised rubbers. The results show that the addition of ZIF-8 to the vulcanisation process leads to an increase in the cross-linking density and an improvement in the mechanical and thermal properties of vulcanised rubbers. The use of ZIF-8 as an activator in combination with traditional activators such as ZnO also gives promising results.

Overall, the paper is organised, well written and fits well with the research theme of the journal. However, there are a few suggestions that could help improve the manuscript.

- As the main contribution of this paper is the use of MOFs as modifiers of the vulcanisation process and rubber properties, it would be more appropriate to discuss this topic in the introduction.

- For clarity, the meaning of all the parameters used should be added to the tables as a footnote.

- Some typos should be eliminated.

Comments on the Quality of English Language

Minor editing of English language required

Author Response

Thank you very much for reviewing and providing valuable comments, which contribute to improving the quality of the publication.

Replies to comments included in the review:

  1. As the main contribution of this paper is the use of MOFs as modifiers of the vulcanisation process and rubber properties, it would be more appropriate to discuss this topic in the introduction.

In response to the comment, the section devoted to the literature review was supplemented with information on works devoted to the use of MOFs as modifiers of the vulcanization process and rubber properties. See the corrected manuscript for details.

  1. For clarity, the meaning of all the parameters used should be added to the tables as a footnote.

The information under the tables regarding material symbols and parameters determined has been supplemented.

  1. Some typos should be eliminated.

The manuscript has been thoroughly checked for typos and corrected. 

Reviewer 2 Report

Comments and Suggestions for Authors

This work provides a general review of MOF materials and studies Zn-MOFs for sulfur vulcanization, various characterizations were performed to gain a deeper insight of ZIF-8 as an activator.

Between Line 329-331 and Line 334-336, can the authors provide some explanation as why 25% replacement using ZIF-8 has led to a significant increase of Ea? Whereas a 100% replacement using MOF-5 has led to a significant increase of Ea? Compared to ZnO, what are the roles of ZIF-8 or MOF-5 during the vulcanization?

Are either ZIF-8 or MOF-5 stable at the temperature of vulcanization? During vulcanization, will the Zn2+ be released from the MOF?

Comments on the Quality of English Language

The manuscript was prepared in good quality and only minor editing of English is required.

Author Response

Thank you very much for reviewing and providing valuable substantive comments, which contribute to improving the quality of the publication.

Replies to comments included in the review:

  1. Between Line 329-331 and Line 334-336, can the authors provide some explanation as why 25% replacement using ZIF-8 has led to a significant increase of Ea? Whereas a 100% replacement using MOF-5 has led to a significant increase of Ea? Compared to ZnO, what are the roles of ZIF-8 or MOF-5 during the vulcanization?

In response to the comment, the section devoted to the literature review was supplemented with information on works devoted to the use of MOFs as modifiers of the vulcanization process and rubber properties. See the corrected manuscript for details.

The increase in energy with the addition of 25% of Zn from ZIF-8 may result from the introduction of larger particles into the system, disturbing the operation of ZnO, and due to the relatively small share, their operation as an activator also requires quite large energy inputs. As the amount of ZIF-8 added increases, the vulcanization activation energy decreases. Thus, for the CV_50_ZnO_50_ZIF-8 sample, it is lower than for the reference sample, and the effect increases even more if we take into account the sample containing 75% of zinc from ZIF - 8. For this best sample, the activation energy drops below 20 kJ/mol. Probably, in these systems we are dealing with synergy of action of both activators, which start working relatively quickly, and starting the reaction does not require large energy expenditure.

The role of ZIF-8 and MOF-5 is to activate the vulcanization process by donating zinc.

  1. Are either ZIF-8 or MOF-5 stable at the temperature of vulcanization? During vulcanization, will the Zn2+ be released from the MOF?

From XRD data (see the corrected version of the manuscript for details) neither mixing nor vulcanization changes the structure of MOFs. ToF-SIMS data (see the corrected version of the manuscript for details) indicate the release of zinc ions from the MOFs cage structures. Their participation in the rubber vulcanization process is confirmed by the presence of ions from the reaction products of zinc with stearic acid in the spectra.

Reviewer 3 Report

Comments and Suggestions for Authors

1. The authors have to provide the IR, TGA and SEM for the different Composition (phr) of the rubber mixtures

2.The introduction part of the manuscript needs some re-write up; 3. The title of the article must be revised. 4. The abstract section must be reconstructed to demonstrate the mentality of the study. 5. All the figures could be improved.

6. To further confirm purity, it is necessary to include results of elemental analyses and/or mass spectrometry analyses. Comments on the Quality of English Language

work

Author Response

Thank you very much for reviewing and providing valuable substantive comments, which contribute to improving the quality of the publication.

Replies to comments included in the review:

  1. The authors have to provide the IR, TGA and SEM for the different composition (phr) of the rubber mixtures.

The reviewer is undoubtedly right, however, in the context of achieving the scientific goal of the work, which is to investigate the possibility of using MOFs as coactivators of sulfur vulcanization of SBR, such a wide scope of experimental work does not seem necessary. TGA experiments were performed mainly to check the influence of the curing system modification on the thermal stability and glass transition temperature of rubber vulcanizates. The results of IR and SEM experiments of systems with different activator compositions should be similar, taking into account the fact that the object of the research are unfilled sulfur vulcanizates. In our opinion, it is not necessary to perform such a large number of additional experiments.

  1. The introduction part of the manuscript needs some re-write up.

In response to the comment, the section devoted to the literature review was supplemented with information on works devoted to the use of MOFs as modifiers of the vulcanization process and rubber properties. See the corrected manuscript for details.

  1. The title of the article must be revised.

The title of the article has been corrected taking into account both the review of MOFs and their use in rubber cross-linking (own research). See the corrected manuscript.

  1. The abstract section must be reconstructed to demonstrate the mentality of the study.

The abstract of the article has been supplemented to show the mentality of the conducted research. See the corrected manuscript for details.

  1. All the figures could be improved.

We tried our best to meet the expectations of the Reviewer.

  1. To further confirm purity, it is necessary to include results of elemental analyses and/or mass spectrometry analyses.

We used commercial MOFs for the research. They have been characterized by the manufacturers, therefore we do not see the need to verify the information provided. Characterizing the purity of MOFs would be justified if we synthesized them ourselves. The information about the compounds used has been completed. See the section 3.1 for details.

Round 2

Reviewer 3 Report

Comments and Suggestions for Authors

accept